# Prognostic Value of a New Right Ventricular-to-Pulmonary Artery Coupling Parameter Using Right Ventricular Longitudinal Shortening Fraction in Patients Undergoing Transcatheter Aortic Valve Replacement: A Prospective Echocardiography Study

**DOI:** 10.3390/jcm13041006

**Published:** 2024-02-09

**Authors:** Christophe Beyls, Mathilde Yakoub-Agha, Alexis Hermida, Nicolas Martin, Maxime Crombet, Thomas Hanquiez, Alexandre Fournier, Geneviève Jarry, Dorothée Malaquin, Audrey Michaud, Osama Abou-Arab, Laurent Leborgne, Yazine Mahjoub

**Affiliations:** 1Department of Anesthesiology and Critical Care Medicine, Amiens University Hospital, F-80054 Amiens, France; yakoubagha.mathilde@chu-amiens.fr (M.Y.-A.); crombet.maxime@chu-amiens.fr (M.C.); abouarab.osama@chu-amiens.fr (O.A.-A.); mahjoub.yazine@chu-amiens.fr (Y.M.); 2UR UPJV 758 SSPC (Simplification of Care of Complex Surgical Patients) Research Unit, University of Picardie Jules Verne, F-80054 Amiens, France; 3Rythmology Unit, Department of Cardiology, Amiens University Hospital, F-80054 Amiens, France; hermida.alexis@chu-amiens.fr; 4Cardiac Intensive Care Unit, Department of Cardiology, Amiens University Hospital, F-80054 Amiens, France; martin.nicolas@chu-amiens.fr (N.M.); hanquiez.thomas@chu-amiens.fr (T.H.); jarry.genevièmalaquin.dorothee@chu-amiens.fr (D.M.); leborgne.laurent@chu-amiens.fr (L.L.); 5Biostatistics Unit, Clinical Research and Innovation Directorate, Amiens-Picardie University Hospital Centre, F-80054 Amiens, France

**Keywords:** right ventricular, coupling, longitudinal shortening fraction, speckle tracking

## Abstract

**Introduction**: Right-ventricular-to-pulmonary artery (RV-PA) coupling, measured as the ratio of tricuspid annular plane systolic excursion (TAPSE) to pulmonary artery systolic pressure (PASP), has emerged as a predictor factor in patients undergoing transcatheter aortic valvular replacement (TAVR). Right ventricular longitudinal shortening fraction (RV-LSF) outperformed TAPSE as a prognostic parameter in several diseases. We aimed to compare the prognostic ability of two RV-PA coupling parameters (TAPSE/PASP and the RV-LSF/PASP ratio) in identifying MACE occurrences. **Method**: A prospective and single-center study involving 197 patients who underwent TAVR was conducted. MACE (heart failure, myocardial infarction, stroke, and death within six months) constituted the primary outcome. ROC curve analysis determined cutoff values for RV-PA ratios. Multivariable Cox regression analysis explored the association between RV-PA ratios and MACE. **Results:** Forty-six patients (23%) experienced the primary outcome. No significant difference in ROC curve analysis was found (RV-LSF/PASP with AUC = 0.67, 95%CI = [0.58–0.77] vs. TAPSE/PASP with AUC = 0.62, 95%CI = [0.49–0.69]; *p* = 0.16). RV-LSF/PASP < 0.30%.mmHg^−1^ was independently associated with the primary outcome. The 6-month cumulative risk of MACE was 59% (95%CI = [38–74]) for patients with RV-LSF/PASP < 0.30%.mmHg^−1^ and 17% (95%CI = [12–23]) for those with RV-LSF/PASP ≥ 0.30%.mmHg^−1^; (*p* < 0.0001). **Conclusions:** In a contemporary cohort of patients undergoing TAVR, RV-PA uncoupling defined by an RV-LSF/PASP < 0.30%.mmHg^−1^ was associated with MACE at 6 months.

## 1. Introduction

Right ventricular (RV) systolic dysfunction (RVsD) and pulmonary hypertension are well-established prognostic factors for major adverse cardiovascular events (MACE) in patients with severe aortic stenosis (AS) undergoing transcatheter aortic valve replacement (TAVR) [1]. A novel non-invasive echocardiographic parameter, the RV-pulmonary artery (RV-PA) coupling parameter, has emerged as an integrative measure of RV performance under varying afterload conditions [2]. This parameter, gauged by the ratio of tricuspid annular plane systolic excursion (TAPSE) to pulmonary artery systolic pressure (PASP), encapsulates the interplay between RV contractile function and pulmonary hemodynamics, thus offering a nuanced assessment of RV adaptability in the context of severe AS [3]. Recent studies in initial cohorts of TAVR patients have shown the potential value of this parameter by revealing a significant association between baseline TAPSE/PASP ratio and outcomes [4,5].

However, two-dimensional speckle tracking parameters (2D-STE) appear more reliable than TAPSE in RVsD identification [6,7]. The right ventricular longitudinal shortening fraction (RV-LSF) is a rapid and reliable 2D-STE parameter to assess global RV systolic function [8]. It has demonstrated superior performance to TAPSE in cardiovascular disease [9,10], especially in patients with pulmonary hypertension [11]. Currently, there is limited data regarding using RV-LSF/PASP as an RV-PA coupling parameter in contemporary TAVR patients.

We hypothesized that the RV-LSF/PASP ratio has better accuracy than the TAPSE/PASP ratio in identifying high-risk patients for MACE occurrence and was associated with poor outcomes. Consequently, we compared the predictive effectiveness between the RV-LSF/PASP ratio and the TAPSE/PASP ratio in detecting MACE within a contemporary cohort of TAVR patients. Furthermore, we evaluated the prognostic relevance of both RV-PA coupling parameters in this patient population.

## 2. Material and Methods

### 2.1. Population

This prospective single-center cohort study involved patients scheduled for TAVR procedures performed by the Heart Team at our University Hospital for severe AS and hospitalized in the Cardiovascular Intensive Care Unit (CICU) at the same hospital.

Inclusion criteria were as follows: Adult patients (>18 years of age) for scheduled TAVR due to severe AS and undergoing the procedure by the Heart Team of our hospital. Patients had to undergo transthoracic echocardiography (TTE) the day before the TAVR with high-quality images suitable for assessing RV-PA coupling parameters, including RV-LSF/PASP and TAPSE/PASP ratios. Patients were included on the day of the TAVR procedure.

Exclusion criteria were patients with poor image quality for RV-PA coupling analysis and PASP measurement, rapid (>110 bpm) arrhythmia during TTE, ventricular pacing, and patients who died during the TAVR procedure.

### 2.2. Outcome

The primary outcome was the ability of RV-PA coupling parameters (RV-LSF/PASP and TAPSE/PASP) to identify patients who experienced MACE during the 6-month follow-up. The secondary outcome was to assess the association between RV-PA coupling parameters and the occurrence of MACE. MACE was defined as all-cause death, myocardial infarction, hospitalization, stroke, and emergency consultation for acute cardiac failure.

### 2.3. Data

Clinical, biological, echocardiographic, and demographic data for each patient during the TAVR procedure and CICU hospitalization were collected prospectively using electronic medical records. Demographic information included age, sex, and body mass index. Cardiovascular risk factors such as hypertension, dyslipidemia, active smoking, and diabetes were documented. Medical history included ischemic heart disease, coronary artery disease, angioplasty, peripheral vascular disease, history of cardiac surgery or MI, sleep apnea, and chronic renal failure were collected. EuroSCORE II and Charlson Score were calculated before the TAVR procedure. Clinical, biological, and echocardiographic data collected during the TAVR procedure are detailed in the Appendix A. Follow-up data and MACE criteria were obtained from phone calls and the patient’s electronic medical records six months after the TAVR procedure.

### 2.4. Echocardiography and RV Systolic Function

Trained operators conducted transthoracic echocardiograms (TTE) the day before the TAVR procedure. Echocardiograms were performed following a standardized protocol based on international guidelines (3)(15) (http://dx.doi.org/10.1016/j.echo.2014.10.003). High-quality echocardiographic images were acquired using a commercially available ultrasound system (CX 50, Philips Healthcare, Andover, MA, USA). Three consecutive heart cycles were recorded and averaged for patients in sinus rhythm, while five cardiac cycles were averaged for those in atrial fibrillation.

### 2.5. RV Systolic Dysfunction

Assessment of RV systolic function using conventional parameters (TAPSE, RV-S’, and RV fractional area change (RV-FAC)) was performed following international guidelines. Thresholds for RV systolic dysfunction (RVsD) were also determined by international guidelines.

RVsD was defined by a RV-LSF < 20% according to recent published studies [8,10].

### 2.6. RV Coupling Parameters Measurement

TAPSE/PASP and RV-LSF/PASP (Figure 1) ratios were measured as follows:(1)According to guidelines [12], TAPSE was measured using M-mode with a cursor positioned at the junction of the lateral tricuspid leaflet and the RV-free wall.(2)A published report described the methodology for RV-LSF analysis [13]. To analyze RV-LSF, three points were used to initialize the first diastolic frame in an apical four-chamber view. These points were placed as follows: (1) at the tricuspid annulus, specifically at the insertion of the anterior tricuspid valve leaflet (RV free wall), (2) at the tricuspid annulus, specifically at the insertion of the septal leaflet, and (3) at the RV apex. The software used for analysis was Automated Cardiac Motion Quantification (QLAB version 9.0, Philips Medical Systems, Andover, MA, USA). The software automatically tracked and calculated the RV-LSF.(3)PASP was calculated using the maximal tricuspid regurgitant jet velocity obtained from continuous wave Doppler and integrated into the modified Bernoulli equation, plus the right atrial pressure. Right atrial pressure was estimated based on the inferior vena cava size (normal ≤ 2.1 cm) and variability with respiration (>50% diameter change with inspiration), following international guidelines [14].

## 3. Statistical Analysis

Data are presented as mean ± standard deviation (SD), median [interquartile range], or numbers (percentage), as appropriate. A receiver-operating characteristic curve (ROC) was constructed to assess the diagnostic performance of the TAPSE/PASP ratio and RV-LSF/PASP ratio in identifying patients experiencing 6-month MACE. The areas under the ROC curves (AUC) for echocardiographic parameters were compared using the DeLong test. After determining the threshold values for RV-PA coupling parameters, the population was dichotomized into MACE and no MACE groups. Variables were compared between the groups using Mann–Whitney or Chi-square tests, as appropriate.

Univariable and multivariable Cox models were conducted to evaluate independent factors associated with the occurrence of MACE. The EuroScore II and Charlson scores were adjusted to identify individuals at high risk of MACE. The median BNP value in the study’s general population was selected as the threshold.

We included significant univariable (*p* < 0.05) in a baseline multivariable proportional hazard ratio (HR) model (Model A), which included conventional RVsD parameters defined according to international guidelines. We then introduced each RV-PA coupling parameter separately: TAPSE/PASP ratio (Model B) and RV-LSF/PASP ratio (Model C). We verified the assumption of proportional HR. The additional predictive value of the TAPSE/PASP ratio and the RV-LSF/PASP ratio was assessed using Harrell’s C-statistic increment. The predictive ability of the models was evaluated based on the Akaike information criteria (AIC). The best model was selected based on the AIC closest to zero [15]. For each RV-PA coupling parameter, cumulative risk curves were generated using the Kaplan–Meier method and compared by the log-rank test.

All tests were two-sided, and the threshold for statistical significance was set to *p*  <  0.05. Statistical analysis was performed with R studio software for macOS (version 2021.09.1 +372) and its «dplyr», «ggplot2», «survminer», «survival», and «compareGroups» packages.

## 4. Results

During the study period (1 January 2021 to 12 December 2022), 446 patients underwent scheduled TAVR. Among them, 239 patients met the inclusion criteria, and 42 were excluded (see Appendix A). Finally, 197 patients were dichotomized into two groups according to the presence or absence of MACE during the 6-month follow-up: 46 patients (23%) in the MACE group and 151 patients (77%) in the no MACE group. The primary cause of MACE was heart failure necessitating hospitalization (n = 37; 80%), followed by all-cause mortality (n = 7; 16%) and, finally, stroke (n = 2; 4%).

Characteristics of the population before TAVR are summarized in Table 1. Euroscore 2 and BNP were higher in the MACE group than in the no MACE group (8.1 [4.4–11.2]% vs. 5.3 [3.6–7.9]%; *p* = 0.016 and 310 [187–704] ng.L^−1^ vs. 191 [122–389] ng.L^−1^; *p* = 0.011, respectively). 

Regarding echocardiographic parameters (Table 2), we found no significant difference in RV size, and no patient exhibited tricuspid regurgitation greater than grade 2. Regarding RV systolic function parameters, the RV-LSF was significantly more impaired in the MACE group compared to the no MACE group with values of 17.1 [12.4–20.4]% vs. 21.0 [16.6–24.1]%; *p* < 0.001. Besides, RV-PA coupling parameters were markedly impaired in the MACE group compared to the no MACE group (TAPSE/PASP at 0.48 [0.32–0.69] mm.mmHg^−1^ vs. 0.62 [0.40–0.76] mm.mmHg^−1^; *p* = 0.049, and RV-LSF at 0.48 [0.25–0.59]%.mmHg^−1^ vs. 0.57 [0.39–0.74]%.mmHg^−1^; *p* = 0.002, respectively). After TAVR, only the occurrence of new onset of atrial fibrillation was significantly higher in the MACE group than the no MACE group (n = 12/46, 26% vs. n = 12/151, 8%, *p* < 0.001).

### 4.1. ROC Curve Analysis

ROC curve analysis has shown that RV-PA coupling parameters are underperforming in identifying patients who experienced 6-month MACE (all AUC < 0.7). The comparison of AUC values revealed no significant difference between the parameters (AUC = 0.67, 95%CI = [0.58–0.77] vs. 0.62, 95%CI = [0.49–0.69]; *p* = 0.16, respectively). The optimal threshold for identifying patients with a 6-month MACE event using RV-LSF/PASP was 0.30%.mmHg^−1^, resulting in a sensitivity of 86% and a specificity of 43%. In contrast, the TAPSE/PASP ratio’s threshold value was 0.55 mm.mmHg^−1^ (see Figure 2).

### 4.2. Univariable Analysis

In univariable logistic regression analysis, several parameters were associated with an increased risk 6 months post-TAVR MACE, as described in Table 3. In the pre-TAVI data, Euroscore II > 7 and BNP levels >200 ng/L were found to have an HR of 2.34, 95%CI = [1.27–4.28]; *p* = 0.010, and 3.53, 95%CI = [1.72–7.29]; *p* = 0.001, respectively.

In the realm of echocardiographic data analysis, the presence of RV systolic dysfunction, as defined by an RV-S’ wave velocity < 9.5 cm/s (HR at 2.21, 95%CI = [1.12–4.04], *p* = 0.02) and an RV-LSF < 20% (HR at 2.90, 95%CI = [1.50–5.70], *p* < 0.001), was observed to be significantly associated with the occurrence of MACE at six months. The RV-LSF/PASP ratio < 0.30%.mmHg^−1^ was the only RV-PA coupling parameter found to be significantly associated with the occurrence of MACE, displaying an HR of 3.97 (95%CI = [2.11–7.49]%.mmHg^−1^; *p* = 0.001). Among post-TAVI complications, only the occurrence of the onset of atrial fibrillation was associated, with an HR of 2.70, 95%CI = [1.33–5.60], *p* = 0.006 (Table 3).

### 4.3. Multivariable Analysis

We developed multiple models incorporating variables identified in the univariable analysis to investigate the relationship between RV-PA coupling parameters and MACE and the additional value of these parameters. In Model A (Table 3), a BNP level > 200 ng/L (HR = 2.20, 95CI% = [1.16–4.10]), the new onset of atrial fibrillation (HR = 2.70, 95%CI = [1.33–5.60]), and the RV systolic dysfunction, defined by an RV-LSF < 20% (HR = 2.80, 95%CI = [1.41–5.50]), were independently associated with MACE (all *p* < 0.05). The AIC of model A was 452, and the C-index was 0.74.

We incorporated each RV-PA coupling parameter into Model A one at a time. In contrast to the TAPSE/PASP ratio (*p* = 0.45), the RV-LSF/PASP ratio was found to be independently associated with MACE (HR = 2.40, 95%CI = [1.20–5.00], *p* = 0.022), making Model C the best performing (AIC at 448 and C-index at 0.76). In Model C, the RvsD, defined by the RV-LSF < 20%, remained independently associated with MACE (see Appendix A).

The 6-month cumulative risk of MACE was 59% (95%CI at [38–74]) with RV-LSF/PASP < 0.30%.mmHg^−1^ and 17% (95%CI at [12–23]) with RV-LSF/PASP ≥ 0.30%.mmHg^−1^ (*p* < 0.0001, Figure 3).

## 5. Discussion

The results of our study, which evaluated the prognostic capability of RV-PA coupling parameters, specifically the TAPSE/PASP and RV-LSF/PASP ratios, in identifying patients with MACE within a contemporary cohort of patients undergoing TAVR, can be summarized as follows: (1) In ROC curve analysis, TAPSE/PASP and RV-LSF/PASP ratio were not optimal for identifying patients who had experienced a MACE; (2), among the two RV-AP coupling parameters, only the RV-LSF/PASP ratio before TAVR was independently associated with MACE and (3) a RVsD before TAVR, defined by a RV-LSF < 20%, was independently associated with MACE.

### 5.1. Pre-TAVR TAPSE/PASP and MACE

The assessment of RV-PA coupling parameters revealed that neither the RV-LSF/PASP ratio nor the TAPSE/PASP ratio could identify patients at risk of MACE with AUC values below 0.7. Besides, a TAPSE/PASP ratio < 0.55 was not associated with MACE. This cutoff value is identical to that proposed by recent publications to define RV-PA decoupling [3]. Our findings align with recent research emphasizing the limited clinical relevance of TAPSE/PASP parameters before TAVR in identifying patients at risk of MACE.

In the bicentric study by Meucci involving 900 patients, the pre-TAVR TAPSE/PASP ratio was not significantly associated with all-cause mortality (adjusted HR = 0.88, 95%CI = [0.45–1.72], *p* = 0.71), even among patients exhibiting severe uncoupling (adjusted HR = 0.85, 95%CI = [0.71–1.29], *p* = 0.77) [5]. Moreover, a study conducted by Parasca et al., which investigated the impact of TAPSE/PASP in 160 patients with severe AS before TAVR, reported a modest AUC of 0.45 for TAPSE/PASP and no significant associations were observed between TAPSE/PASP and MACE (HR = 0.26, 95%CI = [0.66–1.06]) or mortality (HR = 0.50, 95%CI = [0.10–2.55]) [7]. 

RV function and pulmonary hypertension independently predict mortality in patients undergoing TAVR. Given their clinical significance, the simultaneous assessment of TAPSE, a widely adopted, rapid, and reproducible parameter in routine clinical practice, and PASP appeared more compelling than evaluating these parameters separately. However, despite encouraging findings in some studies, the clinical applicability of RV-PA coupling for risk stratification of MACE before TAVR remains uncertain [5,7]. This uncertainty may be due to potential limitations associated with TAPSE, which predominantly assesses longitudinal RV systolic function. Assuming that TAPSE can evaluate RV global systolic function is an overestimation of its diagnostic ability. Moreover, changes in afterload [13] and subsequent progression of RV impairment in AS frequently lead to stages marked by RV dilatation [16] and remodeling, which could challenge the reliability and accuracy of TAPSE [17].

### 5.2. RV-LSF/PASP and MACE

Nevertheless, our results reinforce that speckle-tracking-based parameters are more efficient in assessing high-risk MACE patients after TAVR. In our study, RV-LSF/PASP before TAVR was independently associated with MACE, contrary to TAPSE/PASP. To date, this is the first study that used this parameter to evaluate TAVR patients. These findings aligned with recent research papers, highlighting the improved accuracy and usefulness of RV-LSF as a 2D-STE parameter for assessing RV systolic function, especially in clinical situations involving RV preload or afterload changes [9,10,13]. The demonstrated superiority of speckle tracking-based parameters to evaluate RV function is well-established [17]. In the context of TAVR, our findings are in line with those of the study of Parasca et al., which showed that RV-GLS, another 2D-STE parameter using strain analysis, outperformed the TAPSE/PASP ratio in identifying patients with high-risk mortality after TAVR and was associated with MACE contrary to TAPSE/PASP [7].

### 5.3. RVsD Defined by the RV-LSF and MACE

In our study, RVsD before TAVR, defined by a RV-LSF < 20%, was the only RV systolic function parameter independently associated with MACE at six months. This finding is consistent with several studies demonstrating an association between RVsD, assessed through an RV-LSF < 20%, and the occurrence of MACE [9,10]. Indeed, RV-LSF seems to be a more precise than conventional parameters for identifying patients with RVsD in various cardiovascular pathologies [8,11]. RV-LSF assesses the global RV systolic function, contrary to TAPSE, and likely enables early detection of RVsD [18]. In our study, patients did not exhibit significant impacts on the size of the RV chamber, including the absence of major tricuspid regurgitation or significant pulmonary hypertension. This is probably because most patients presented with normal-flow high-gradient AS receiving treatment for their AS at an earlier stage, and medical optimization likely helped prevent RV impairment, which is a crucial step in the progression of AS disease.

### 5.4. Clinical Value of RV-PA Coupling Parameters

Despite the promising potential of RV-LSF/PASP, the role of RV-PA coupling assessment in TAVR and its clinical usefulness remain to be defined. In our study, even though the RV-LSF/PASP ratio is associated with MACE, its added value is limited compared to other well-established prognostic factors, such as the EuroScore II or BNP. Positive results from multicentric studies involving contemporary cohorts are necessary to recommend the routine of RV-PA coupling before TAVR for cardiovascular risk stratification. However, several studies suggested that monitoring RV-PA coupling after TAVR could help identify patients with fewer benefits from TAVI, thereby highlighting the potential significance of this promising parameter [3,5].

### 5.5. Strengths and Limits

One of the notable strengths of our study lies in the timing of TTE examinations, which were performed one day before TAVR, ensuring a precise assessment of RV systolic function and PASP. Additionally, our findings hold relevance for contemporary patients, as we conducted this study on a cohort of TAVR patients managed by a highly skilled team utilizing third-generation valves. This approach resulted in fewer post-procedure paravalvular leaks and a consistent incidence of ischemic strokes, aligning with modern TAVR practices.

Nevertheless, we acknowledge several limitations in our research. Firstly, only TAVR procedures performed by the Heart Team at our University Hospital were included. Indeed, we are the sole tertiary center with authorization for TAVR procedures, while two other teams conducted TAVR procedures at our center. We excluded 149 patients from these teams because they have their own patient selection and follow-up processes, making it challenging to monitor patients post-procedure and collect data.

Secondly, our sample size did not allow for an analysis of different RV-PA ratios, including quartiles, and we did not calculate the required number of patients because we did not have data on RV-LSF in this population. Moreover, our cohort was not originally designed or powered to evaluate clinical outcomes associated with RV-PA coupling parameters. However, if this study had been conducted prospectively, to achieve a 15% difference in sensitivity and specificity with a 0.05% bilateral alpha risk (which is an acceptable limit for comparing diagnostic tests), it would have been necessary to include 179 patients. Given the relatively low incidence of MACE in contemporary TAVI with third-generation valves, recruiting a larger patient population would have been necessary to achieve sufficient statistical power for discerning meaningful clinical outcomes among different coupling levels.

In our study, the 6-month incidence of MACE was 23% (n = 46/197), notably higher than the rates reported in international multicenter randomized trials involving patients with intermediate risk. For instance, in the PARTNER 2 study, only 13.9% of TAVR patients experienced a MACE (comprising hospitalization, myocardial infarction, stroke, or all-cause death) within 30 days [19]. Similarly, the SURTAVI randomized trial reported a 13.2% incidence of major cardiovascular events among TAVR patients at 12 months. The variation in outcomes can be ascribed to the real-world nature of our cohort, which included a high-risk population [20].

The RV-FLWS/PASP ratio is attractive for assessing RV-PA coupling [7]. However, our study did not include RV-FLWS in the statistical analysis because more than 20% of patients with an RV-LSF measurement could not obtain a reliable RV-FLWS measurement using Autostrain software.

Finally, it is essential to note that, like many 2D-STE parameters, the RV-LSF value is influenced by the software used and its version. To ensure consistency in monitoring RV systolic function, it is recommended to use the same software throughout follow-up [21].

## 6. Conclusions

In a cohort of contemporary real-life patients with aortic stenosis undergoing TAVR, the pre-procedure RV-PA coupling parameter defined by the RV-LSF/PASP ratio was associated with MACE contrary to the conventional TAPSE/PASP ratio. These findings need confirmation in multicentric investigations.

## Figures and Tables

**Figure 1 jcm-13-01006-f001:**
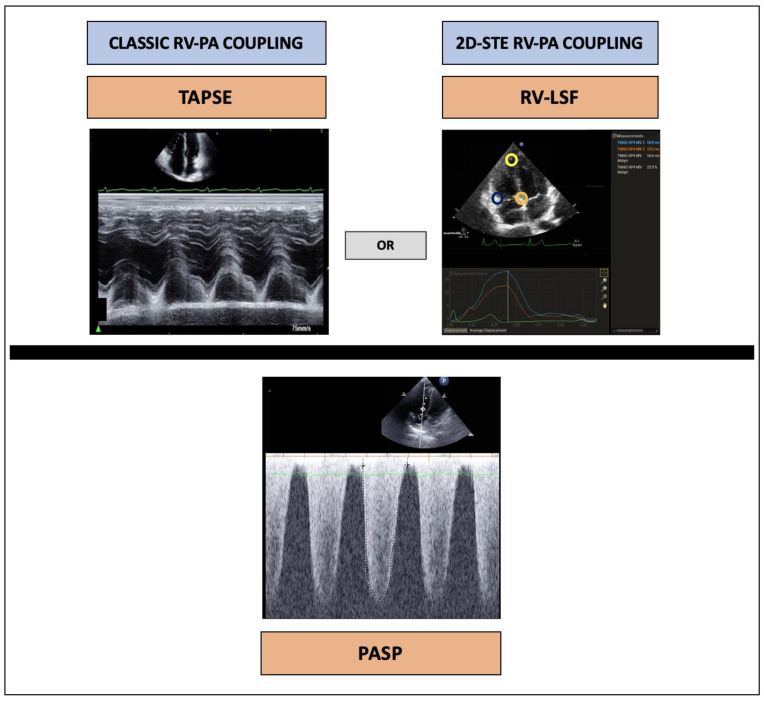
Measurement of RV-PA parameter in echocardiography: conventional (TAPSE/PASP) and 2D-STE (RV-LSF/PASP) parameters. PASP: pulmonary artery systolic pressure; RV-PA: right ventricular to pulmonary artery; RV-LSF: right ventricular longitudinal shortening fraction; TAPSE: tricuspid annular plane systolic excursion.

**Figure 2 jcm-13-01006-f002:**
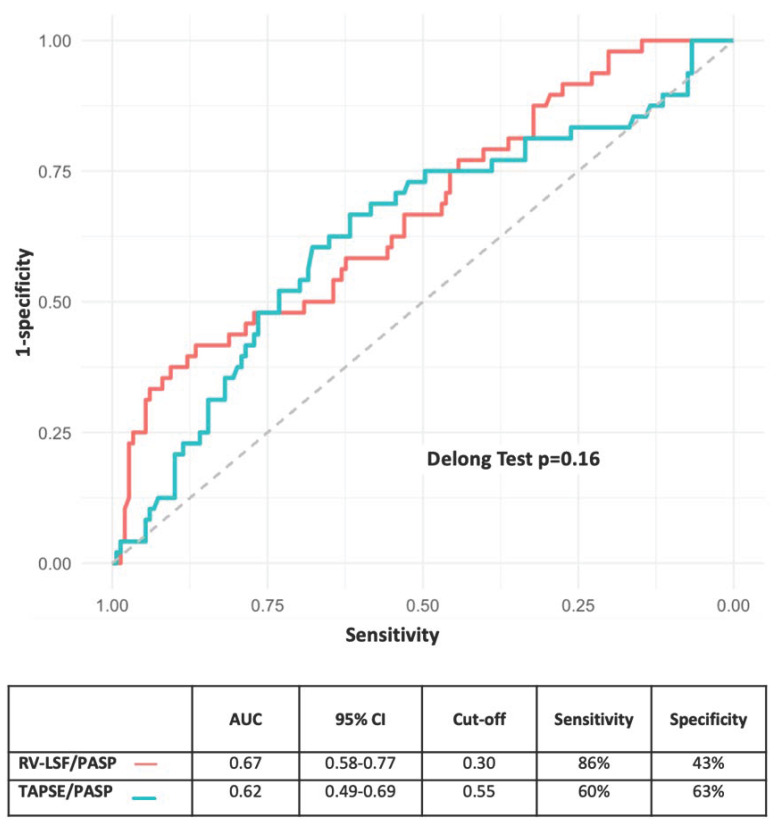
ROC curve analysis of RV-PA parameters for prediction MACE in patients with severe AS undergoing TAVR. AS: aortic stenosis; MACE: major cardiovascular clinical event; RV-PA: right ventricular to pulmonary artery; TAVR: transcatheter aortic ventricular replacement.

**Figure 3 jcm-13-01006-f003:**
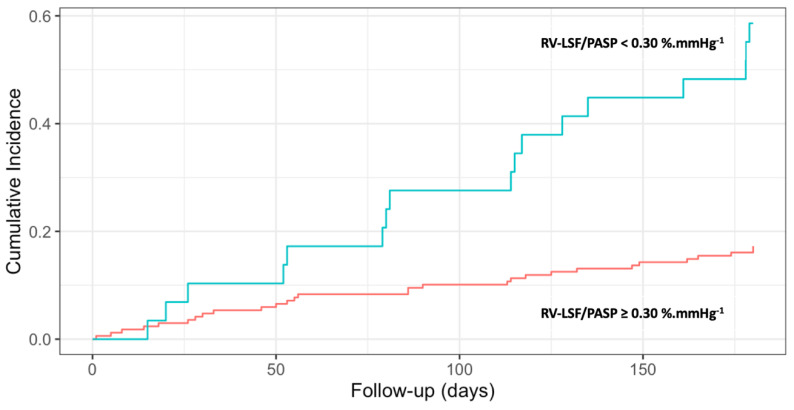
Cumulative risk of MACE according to RV-LSF/PASP ratio. MACE: major cardiovascular clinical event; PASP: pulmonary artery systolic pressure; RV-LSF: right ventricular longitudinal shortening fraction.

**Table 1 jcm-13-01006-t001:** Demographic data of the studied population.

Variables	No MACE N = 151	MACE N = 46	*p* Value
Age, years	81 [76–85]	82 [76–86]	0.49
BMI, kg.m^−2^	27.7 [24.6–31.1]	26.0 [23.3–29.8]	0.09
Female sex, n (%)	60 (39)	20 (45)	0.57
Euroscore II	5.3 [3.6–7.9]	8.1 [4.4–11.2]	0.022
Charlson Score	6.0 [5.0–7.0]	7.5 [5.0–8.0]	0.07
Medical history, n (%)			
Hypertension	126 (83)	40 (87)	0.73
Diabetes mellitus	47 (31)	10 (21)	0.22
Dyslipidemia	73 (48)	22 (48)	1.00
Smoking	46 (30)	14 (30)	1.00
Peripheareal artery disease	11 (7)	8 (18)	0.08
Chronic renal disease	30 (20)	14 (32)	0.13
Cardiac surgery	10 (7)	5 (11)	0.33
Chronic coronary disease	32 (21)	10 (23)	0.96
Myocardial infarction	14 (9)	5 (11)	0.77
Atrial fibrillation	46 (31)	17 (36)	0.90
COPD	19 (12)	10 (23)	0.15
NYHA functional class III or IV	62 (41)	19 (41)	1.00
Permanent pacemaker	15 (10)	5 (11)	0.78
Biology before TAVR			
BNP level, ng/L	191 [122–389]	310 [187–704]	0.011
Calcic score	2822 [2094–4075]	3300 [1661–3897]	0.61
Hemoglobin, g/dL	12.9 [11.8–13.6]	12.4 [11.5–13.7]	0.37
Creatinin, μmol/L	91 [72–111]	96 [75–137]	0.10

Data are expressed as median (25–50) and count (%). BMI: body mass index; COPD: Chronic obstructive pulmonary disease; NYHA: New York Heart Association; BNP: brain natriuretic peptide.

**Table 2 jcm-13-01006-t002:** Echocardiographic data and clinical course after TAVR.

Variables	No MACE N = 151	MACE N = 46	*p* Value
TTE before TAVR			
Aortic valve area (cm^2^)	0.78 [0.65–0.95]	0.77 [0.60–0.92]	0.63
Indexed aortic valve area (cm^2^.m^2^)	0.41 [0.33–0.50]	0.41 [0.33–0.50]	0.93
Aortic peak velocity (m.s^−1^)	4.39 [4.15–4.73]	4.31 [4.03–4.77]	0.14
Mean transaortic gradient (mmHg)	49 [42–56]	46 [41–56]	0.35
E wave (cm.s^−1^)	95 [77–122]	105.5 [82–121]	0.22
A wave (cm.s^−1^)	102 [86–117]	98 [88–114]	0.65
E wave deceleration time (ms)	200 [155–291]	210 [159–262]	0.90
E/A ratio	0.8 [0.7–1.2]	1.1 [0.8–1.4]	0.06
Lateral E	8.4 [6.6–10.2]	7.9 [6.7–10.0]	0.69
LVEF (%)	60 [55–65]	60 [45–65]	0.26
RV parameters			
RV longitudinal dimension (mm)	76 [69–85]	77 [67–86]	0.97
RV mid-cavity dimension (mm)	34 [29–37]	34 [27–36]	0.81
RV basal dimension (mm)	41 [36–45]	41 [34–49]	0.73
RV EDA (cm^2^)	22.0 [18.7–28.0]	23.1 [17.2–27.9]	0.69
RV ESA (cm^2^)	14.4 [10.9–19.3]	15.2 [11.1–18.8]	0.77
Tricuspid peak velocity (cm.s^−1^)	281 [250–310]	280 [255–305]	0.94
PASP (mmHg)	35 [28.2–45]	38 [31.8–45]	0.18
RV systolic function parameters			
TAPSE (mm)	21.5 [18.4–24.8]	19.3 [16.5–24.1]	0.25
RV-S’ (cm.s^−1^)	13 [10.8–14.8]	12.2 [9.8–13.8]	0.06
RV-FAC (%)	39.6 [28.0–56.2]	39.8 [29.6–60.4]	0.38
RV-LSF (%)	21.0 [16.6–24.1]	17.1 [12.4–20.4]	<0.001
RV-PA coupling parameters			
TAPSE/PASP (mm.mmHg^−1^)	0.62 [0.40–0.76]	0.48 [0.32–0.69]	0.049
RV-LSF/PASP (%.mmHg^−1^)	0.57 [0.39–0.74]	0.48 [0.25–0.59]	0.002
Clinical course after TAVR			
Sapien 3	99 (66)	27 (61)	0.70
Corevalve Evolut R	14 (9)	6 (14)	0.40
Corevalve Evolut Pro	33 (22)	9 (21)	1.00
Acurate	2 (2)	2 (4)	0.22
Others	1 (1)	0 (0)	1.00
Access site, n (%)			
Femoral	151 (99)	43 (98)	0.40
Trans-carotid	0 (0)	1 (2)	0.22
Subclavian	1 (1)	0 (0)	1.00
Surgical approach, n (%) *	7 (5)	2 (5)	1.00
Procedure time, min	73 [59–90]	73 [64–93]	0.49
Post procedure complications, n (%)			
Stroke	3 (2)	3 (7)	0.13
Pacemaker implantation	35 (23)	9 (21)	0.95
Acute kidney injury	18 (12)	6 (14)	0.94
Paravalvular leak			0.78
0	96 (63)	29 (63)	
1	54 (36)	17 (40)	
2	1 (1)	0 (0)	
Transfusion	12 (8)	6 (14)	0.24
New onset of atrial fibrillation	12 (8)	12 (26)	0.001

* The surgical approach was defined by the fact that a surgeon had to perform surgical access at the femoral site in case of a failure to puncture the femoral vessels or had to pre-dilate the vessels before the TAVR team could proceed with their procedure. EDA: end-diastole area; ESA: end-systole area; LVEF: left ventricular ejection fraction; PASP: pulmonary arterial systolic pressure; TAPSE: tricuspid annular plane systolic motion; RV: right ventricular; RV-FAC: right ventricle fractional area change; RV-LSF: right ventricle longitudinal shortening fraction.

**Table 3 jcm-13-01006-t003:** Cox Analysis of Predictive Models for Association with MACE.

(A) Univariable and Multivariable Cox analysis of variables associated with MACE at 6 months.				
	Univariable analysis		Multivariable Model A	
	HR (95%CI)	*p*	HR (95%CI)	*p*
Euroscore II > 7	2.34 [1.27–4.28]	0.011	1.40 [0.76–2.70]	0.26
Charlson score > 6	1.72 [0.95–3.13]	0.08	-	-
Critical aortic stenosis	0.34 [0.20–2.51]	0.29	-	-
BNP > 200 ng.L^−1^	3.53 [1.72–7.29]	0.001	2.20 [1.16–4.10]	0.02
New-onset atrial fibrillation	2.16 [0.83–5.61]	0.12	2.70 [1.33–5.60]	0.006
Pacemaker post TAVR	0.76 [0.35–1.66]	0.51	-	-
Paravalvular leak > grade 1	1.11 [0.60–2.01]	0.75	-	-
PASP > 45 mmHg^−1^	1.50 [0.76–2.80]	0.25		
Tricuspid regurgitation vmax > 2.9 m.s^−1^	0.99 [0.55–1.80]	0.97		
Right ventricular dysfunction				
- RV-S’ < 9.5 cm/s	2.21 [1.12–4.04]	0.024	1.30 [0.65–2.50]	0.48
- TAPSE < 17 mm	1.80 [0.94–3.49]	0.08	-	-
- RV-FAC < 35%	0.74 [0.41–1.35]	0.31	-	-
- RV-LSF < 20%	2.90 [1.50–5.70]	<0.001	2.80 [1.41–5.50]	0.003
RV-PA coupling				
- TAPSE/PAPS < 0.55	1.80 [0.98–3.34]	0.06	NR	-
- RV-LSF/PASP < 0.30	3.97 [2.11–7.49]	0.001	NR	-
(B) Discrimination of Cox model according to RV-PA coupling parameter				
	AIC	C index	HR of the RV-PA coupling parameter	*p*
Multivariable Model A	452	0.74	-	-
Model B: Multivariable Model A + TAPSE/PASP < 0.55 mm.Hg^−1^	454	0.74	1.30 [0.67–2.40]	0.45
Model C: Multivariable Model A + RV-LSF/PASP < 0.30%.mmHg^−1^	44!	0.76	2.40 [1.20–5.0]	0.022

NR: not retained for the model A analysis. AIC: aikake information criteria; CI: confidence interval; HR: hazard ration; RV-FAC: right ventricular fractional area change; RV-LSF: right ventricular longitudinal shortening fraction; RV-PA: right ventricular to pulmonary artery; TAPSE: tricuspid annular plane systolic excursion; TAVR: transcatheter valvular replacement.

## Data Availability

The dataset used and analyzed during the current study is available from the corresponding author upon reasonable request.

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
