# Peer review of "Prognostic Value of a New Right Ventricular-to-Pulmonary Artery Coupling Parameter Using Right Ventricular Longitudinal Shortening Fraction in Patients Undergoing Transcatheter Aortic Valve Replacement: A Prospective Echocardiography Study"

_jcm, 2024, doi:10.3390/jcm13041006_

Round 1
Reviewer 1 Report
Comments and Suggestions for Authors
The authors provide a study to investigate if a new right ventricular to pulmonary artery coupling parameter using right ventricular longitudinal shortening fraction instead of TAPSE has a higher prognostic value for MACE at 6 months in patients undergoing TAVR. None of the ratios (TAPSE/ PASP ratio and RV-LSF/ PASP) demonstrated a convincing AUC in ROC analysis (both <0.70) and showed no significant difference. Further, the authors provided cut-off values for both parameters showing their novel parameter (RV-LSF/ PASP ratio <0.30 %.mmHg-1) to be independently associated with MACE at 6 months while TAPSE/ PASP ratio did not.
Although the concept of RV-PA coupling is attracting attention in many clinical conditions, I have several comments as listed below, which limit the quality of the manuscript presented:
Comments
1. The authors need to clarify the size of the study population as well as the number of patients not meeting inclusion criteria or meeting exclusion criteria as patient numbers are different in the manuscript and the supplemental material. Also clarify the statement on 475 patients underwent scheduled TAVR (stated in the manuscript) but many patients were excluded for TAVR not performed (stated in supplemental material).
2. The method used for the quantification of RV-LSF is a three-point approach mostly depending on the movement of the point set on the insertion point of the anterior tricuspid leaflet at the tricuspid annulus. As both other points, which are set at the apex and the interventricular septum, are less mobile, this approach is resembling TAPSE. This is supported by the observation of the authors that no difference in ROC curve analysis was found for RV-LSF/ PASP and TAPSE/ PASP ratio. Furthermore, the cited references underscoring the value of this approach to assess RV-LSF originate from the same first author (references 9, 10, 13). A more comprehensive quantification for RV-LSF is RV free wall strain, which has the potential to overcome this methodological limitation and is also cited as a well-established standard by the authors (see reference 17, or doi: 10.1016/j.jacc.2023.05.010.).
3. Can the authors provide information on the investigation of association of RV-LSF, TAPSE and PASP alone on MACE occurrence at 6 months?
4. Can the authors please provide information on the distribution of MACE as the authors observed a strong independent association of BNP and new onset atrial fibrillation suggesting high rates of heart failure hospitalization. Is it possible to identify a BNP cut-off independently associated with MACE at 6 months?
5. I am curious about the defined exclusion cut-off >140 bpm during TTE as heart rate significantly influences ventricular function. Current ESC guidelines (https://doi.org/10.1093/eurheartj/ehaa612) recommend a rate control <110 bpm (during moderate exercise), which might be a more reasonable cut-off.
6. Several clinical or demographic data suggest a more high-risk collective in the MACE cohort irrespective of echocardiographic data such as a significantly higher EURO-Score II, a significantly higher BNP level and a higher proportion of patients exhibiting new onset atrial fibrillation post TAVR. With these well-established parameters Model A ended up with AIC 418 and C index 0.71. I am questioning the additional value of the new parameter, when Model A + RV-LSF/PAPS <0.30 ended up with AIC 415 and C index 0.74.
7. Please use ‘univariable’ instead of ‘univariate’ and ‘multivariable’ instead of ‘multivariate’ when presenting independent variables as shown in Table 3.
8. No information is given for RV size and the presence/ severity of tricuspid regurgitation, which are important variables when investigating the relationship of RV properties and the occurrence of MACE.
9. Line 195 following: Please clarify the numbers presented as numbers for HR and CI in text and Table 3 vary significantly. The same is true for line 198, in which the data presented for new onset atrial fibrillation is very much different from the data presented in Table 3.
10. Line 199 refers to Table 4, which is not presented in the manuscript.
11. I encourage the authors to use the same format when presenting p-values. Therefore, I suggest three decimal numbers. Please unify data presentation in the entire manuscript.
12. Please check the entire manuscript for missing or additional full stops and missing or additional spaces (i.e. see lines 260, 281, 290).
13. Please check for MACE rate as the given numbers differ in this essential information for the analysis (i.e. see line 291 and 155).
14. Please optimize citation style of reference 18 and 19 (authors missing).
15. Table 2: Please adjust interquartile ranges especially for aortic peak velocity in no MACE cohort.
16. Line 58: reference and full stop is missing, line 60: please add LSF to RV-PASP ratio.
Comments on the Quality of English LanguageOverall English Language is acceptable, but I encourage the authors to have a careful read through and optimized wording and punctuation (i.e. see comments 12 and 16).
Author Response
Reviewer 1.
The authors provide a study to investigate if a new right ventricular to pulmonary artery coupling parameter using right ventricular longitudinal shortening fraction instead of TAPSE has a higher prognostic value for MACE at 6 months in patients undergoing TAVR. None of the ratios (TAPSE/ PASP ratio and RV-LSF/ PASP) demonstrated a convincing AUC in ROC analysis (both <0.70) and showed no significant difference. Further, the authors provided cut-off values for both parameters showing their novel parameter (RV-LSF/ PASP ratio <0.30 %.mmHg-1) to be independently associated with MACE at 6 months while TAPSE/ PASP ratio did not. Although the concept of RV-PA coupling is attracting attention in many clinical conditions, I have several comments as listed below, which limit the quality of the manuscript presented:
Comments
- The authors need to clarify the size of the study population as well as the number of patients not meeting inclusion criteria or meeting exclusion criteria as patient numbers are different in the manuscript and the supplemental material. Also clarify the statement on 475 patients underwent scheduled TAVR (stated in the manuscript) but many patients were excluded for TAVR not performed (stated in supplemental material).
Response: We thank the reviewer for this remark. We apologize for this error and correct the number in the text and the flow chart. We completed the result section, page 12, line 196 as follows : During the study period (January 1, 2021, to December 12th, 2022), 446 patients underwent scheduled TAVR. Among them, 239 patients met the inclusion criteria, and 42 were excluded (see Figure 1 in Supplementary Files). Finally, 197 patients were dichotomized into two groups according to the presence or absence of MACE during the 6-month follow-up: 46 patients (23%) in the MACE group and 151 patients (77%) in the no-MACE group. The primary cause of MACE was heart failure necessitating hospitalization (n=37; 80%), followed by all-cause mortality (n=7; 16%) and, finally, stroke (n=2; 4%).
We added in the limit section the reason of the number of excluded patients, page 19, line 352, as follows: Firstly, only TAVR procedures performed by the Amiens Hospital Team were included. Indeed, we are the sole tertiary center with authorization for TAVR procedures, while two other teams conducted TAVR procedures at our center. We excluded 149 patients from these teams because they have their own patient selection and follow-up processes, making it challenging to monitor patients post-procedure and collect data.
And according to Reviewer 2, we modified the flow chart and added the MACE event (see the new Figure 1 in the supplementary Files)
The method used for the quantification of RV-LSF is a three-point approach mostly depending on the movement of the point set on the insertion point of the anterior tricuspid leaflet at the tricuspid annulus. As both other points, which are set at the apex and the interventricular septum, are less mobile, this approach is resembling TAPSE.
This is supported by the observation of the authors that no difference in ROC curve analysis was found for RV-LSF/ PASP and TAPSE/ PASP ratio.
Furthermore, the cited references underscoring the value of this approach to assess RV-LSF originate from the same first author (references 9, 10, 13).
A more comprehensive quantification for RV-LSF is RV free wall strain, which has the potential to overcome this methodological limitation and is also cited as a well-established standard by the authors (see reference 17, or doi: 10.1016/j.jacc.2023.05.010.).
Response: We thank the reviewer for this remark. We acknowledge the role of longitudinal contraction of the RV free wall, notably in the measurement of RV-LSF, particularly through the measurement of lateral tricuspid annular displacement (TAD lat). However, TAPSE, RV-FLWS, and RV-LSF are different echocardiographic parameters, each with advantages and disadvantages. Firstly, TAPSE and RV-FLWS are more sensitive to loading conditions and only represent the longitudinal function of the right ventricular systolic function. In contrast, RV-LSF is a parameter of the overall right ventricular systolic function. Several studies have demonstrated differences between these parameters, with RV-LSF being more reliable than TAPSE or RV-FLWS in identifying patients with right ventricular dysfunction.
Furthermore, RV-FLWS measurement requires higher-quality images than RV-LSF. In our study, RV-FLWS could only be reliably measured in up to 20% of patients with an RV-LSF measurement, which complicated the statistical analysis. We added this point in the limit section, page 20, line 377, as follows: The RV-FLWS/PASP ratio is attractive for assessing RV-PA coupling (7). However, our study did not include RV-FLWS in the statistical analysis because more than 20% of patients with an RV-LSF measurement could not obtain a reliable RV-FLWS measurement using Autostrain software.
Can the authors provide information on the investigation of association of RV-LSF, TAPSE and PASP alone on MACE occurrence at 6 months?
Response: We thank the reviewer for this remark. We investigate the association of the different RV systolic function parameters (TAPSE, onde S’, RV-FAC, and RV-LSF) and the PASP in univariate Cox analysis. Only the Onde S’ and the RV-LSF were associated with MACE 6 monts. We added the RVLSF into multivariable models (B and C) and changed the HR in table 3.
We change the section results, page 13, line 236: as follows: In the realm of echocardiographic data analysis, the presence of RV systolic dysfunction, as defined by an S' wave velocity < 9.5 cm/s (HR at 2.12 [1.12-4.04], p=0.02), and an RV-LSF <21% (HR at 2.90, 95%CI[1.50-5.70], p<0.001), was observed to be significantly associated with the occurrence of MACE at 6 months. The RV-LSF/PASP ratio < 0.30 %.mmHg-1 was the only RV-PA coupling parameter found to be significantly associated with the occurrence of MACE, displaying a HR of 3.97 (95%CI=[0.02-0.39] %.mmHg-1; p=0.001).
We complete the discussion with this new result into the discussion section, page 17, line 321, as follows: In our study, RVsD before TAVR, defined by a RV-LSF <20%, was the only RV systolic function parameter independently associated with MACE at six months. This finding is consistent with several studies demonstrating an association between RVsD, assessed through an RV-LSF <20%, and the occurrence of MACE(9,10). Indeed, RV-LSF seems to be a more precise parameter than conventional in identifying patients with RVsD in various cardiovascular pathologies (8,11). RV-LSF assesses the global RV systolic function, contrary to TAPSE, and likely enables early detection of RVsD (18). In our study, patients did not yet exhibit significant impacts on the RV, including the absence of major tricuspid regurgitation or significant pulmonary hypertension. This is probably because patients received treatment for their AS at an earlier stage, and medical optimization likely helped prevent RV impairment, which is a crucial step in the progression of AS disease.
- Can the authors please provide information on the distribution of MACE as the authors observed a strong independent association of BNP and new onset atrial fibrillation suggesting high rates of heart failure hospitalization. Is it possible to identify a BNP cut-off independently associated with MACE at 6 months
Response: We thank the reviewer for this remark. The distribution of MACE was provided in the results section on page 12, line 200, as follows: The primary cause of MACE was heart failure necessitating hospitalization (n=37; 80%), followed by all-cause mortality (n=7; 16%) and finally, stroke (n=2; 4%).
We chose the BNP value of 200 because it represents the median value in the study's general population. We conducted an ROC curve analysis for BNP levels to identify patients with MACE during follow-up. The AUC was 0.63 [0.46-0.73], with a threshold value at 183 (sensitivity at 0.76 and specificity at 0.51). We also performed the same statistical test with the threshold value and found no significant difference between the two values.
We explain the choice of this value in the statistical test on page 10, line 176, as follows: The EuroScore II and Charlson scores were adjusted to identify individuals at high risk of MACE. The median BNP value in the study's general population was selected as the threshold. If the reviewer wishes, we can include the new statistical analyses with the threshold of 183 in the supplementary file.
- I am curious about the defined exclusion cut-off >140 bpm during TTE as heart rate significantly influences ventricular function. Current ESC guidelines (https://doi.org/10.1093/eurheartj/ehaa612) recommend a rate control <110 bpm (during moderate exercise), which might be a more reasonable cut-off.
Response: We thank the reviewer for this remark. We agree with the reviewer and have revised the exclusion criteria. With the cut-off lowered to 110, no additional patients were excluded from the analysis. We have corrected page 6, line 107, as follows: Exclusion criteria included patients with poor image quality for RV-PA coupling analysis and PASP measurement, rapid (>110 bpm) arrhythmias during TTE, ventricular pacing, and patients who passed away during the TAVR procedure.
- Several clinical or demographic data suggest a more high-risk collective in the MACE cohort irrespective of echocardiographic data such as a significantly higher EURO-Score II, a significantly higher BNP level and a higher proportion of patients exhibiting new onset atrial fibrillation post TAVR. With these well-established parameters Model A ended up with AIC 418 and C index 0.71. I am questioning the additional value of the new parameter, when Model A + RV-LSF/PAPS <0.30 ended up with AIC 415 and C index 0.74.
Response: We appreciate the reviewer's comment. As suggested, we have included RV-LSF in the new model. The AIC decreased from 452 to 444, and the C-index improved from 0.74 to 0.76. We concur that the added value of the RV-LSF/PAPS ratio is low, and its utility is likely limited. We have tempered our findings in the discussion section on page 17, line 334, as follows: Despite the promising potential of RV-LSF/PASP, the role of RV-PA coupling assessment in TAVR and its clinical usefulness remain to be defined. In our study, even though the RV-LSF/PASP ratio is associated with MACE, its added value is limited compared to other well-established prognostic factors, such as the EuroScore II or BNP.
- Please use ‘univariable’ instead of ‘univariate’ and ‘multivariable’ instead of ‘multivariate’ when presenting independent variables as shown in Table 3.
Response: We thank the reviewer for these remarks, and we have corrected the typographical errors.
- No information is given for RV size and the presence/ severity of tricuspid regurgitation, which are important variables when investigating the relationship of RV properties and the occurrence of MACE.
Response: We thank the reviewer for this remark. We added the RV size dimension in the Table 2. In our cohort, only 1 patient had a mitral regurgitation > grade 2 and was excluded for poor quality imaging. No patient with a TR > grade 2 was included in our study.
We add this result, page 12, line 207, as follows: Regarding echocardiographic parameters (Table 2), we found no significant difference in RV size, and no patient exhibited tricuspid regurgitation greater than grade 2.
In our study, the lack of impact of aortic disease on the right heart is likely attributed to the fact that this is a contemporary TAVR cohort, and the management is probably swifter with more optimized medical treatment. We added this point in the discussion section, page 17, line 327, as follows: In our study, patients did not yet exhibit significant impacts on the RV, including the absence of major tricuspid regurgitation or significant pulmonary hypertension. This is probably because patients received treatment for their AS at an earlier stage, and medical optimization likely helped prevent RV impairment, which is a crucial step in the progression of AS disease.
- Line 195 following: Please clarify the numbers presented as numbers for HR and CI in text and Table 3 vary significantly. The same is true for line 198, in which the data presented for new onset atrial fibrillation is very much different from the data presented in Table 3.
Response: We thank the reviewer for this remark, and we have corrected these typographical errors. We have also updated the HR and CI values in Table 3 with the new data, reflecting the inclusion of RV-LSF in both the univariable and multivariable analysis.
- Line 199 refers to Table 4, which is not presented in the manuscript.
Response: We thank the reviewer for these remarks, and we have corrected the typographical errors.
11.I encourage the authors to use the same format when presenting p-values. Therefore, I suggest three decimal numbers. Please unify data presentation in the entire manuscript.
Response: We thank the reviewer for this remark, and we have corrected the p-value to two decimal places and three decimal places for the most significant variable.
- Please check the entire manuscript for missing or additional full stops and missing or additional spaces (i.e. see lines 260, 281, 290).
- Please check for MACE rate as the given numbers differ in this essential information for the analysis (i.e. see line 291 and 155).
- Please optimize citation style of reference 18 and 19 (authors missing).
- Table 2: Please adjust interquartile ranges especially for aortic peak velocity in no MACE cohort.
- Line 58: reference and full stop is missing, line 60: please add LSF to RV-PASP ratio.
Response: We thank the reviewer for these remarks, and we have corrected the typographical errors.
Reviewer 2 Report
Comments and Suggestions for Authors
This is a study on an interesting topic that investigates a large patient population. It is a small, single-center study with several limitations, which is compensated for by its originality. I find it interesting and believe it could be enhanced with minor changes.
Abstract:
- - The study lacks commentary on the study type in the study methodology.
Introduction:
- - Minor correction: Line 58, what does "(ref)" mean?
- - Otherwise, it is correct, interesting, and introduces the article well. Little to add.
Methodology:
- - Begin by specifying the study type (Is it a cohort study? Is it prospective? Is it single-center?). I understand these are consecutive cases from consecutive patients.
- - If it is prospective (as it appears) but a sample size was not calculated a priori, specify the reason for this and consider calculating the study power post hoc.
- - It seems peculiar that the study's objective was to investigate if the RV-PASP ratio was better than TAPSE/PASP, while all primary and secondary endpoints focused on the RV-PASP relationship and MACE. Comparative data should be included if that is the stated objective.
Results:
- - Over 200 patients who underwent TAVR were excluded from the study. What is the reason? This would mean they are not consecutive cases. This exclusion should be acknowledged as a limitation.
- - It would be beneficial to include MACE events in the patient flowchart, separated by the type of event within this combined outcome.
- - I wonder about the remaining parameters of Speckle Tracking, both left and right ventricular, atrial, and their prognostic value. Were these obtained?
- - Otherwise, it is an interesting and well-structured section.
Discussion:
- - Minor correction: Additional space in line 230, 261, and throughout the section, there is incorrect spacing before and after punctuation marks (line 290). Correct this.
- - It would be interesting to include other right ventricular parameters that have proven useful in the prognostic evaluation of patients undergoing TAVR. It would be relevant to place the studied parameters in this context.
- In limitations, the study's nature, which is not consecutive patients, and the reasons for exclusion should also be included.
Author Response
Reviewer 2
Abstract:
- The study lacks commentary on the study type in the study methodology.
Response: We thank the reviewer for this comment and complete the methodology section of the abstract, page 3, line 48, as follows: A prospective and single-center study involving 197 patients who underwent TAVR at Amiens University Hospital.
Introduction:
- Minor correction: Line 58, what does "(ref)" mean?
Response: We apologize and we correct this typo error
- Otherwise, it is correct, interesting, and introduces the article well. Little to add.
Methodology:
- Begin by specifying the study type (Is it a cohort study? Is it prospective? Is it single-center?). I understand these are consecutive cases from consecutive patients.
Response: We thank the reviewer for this remark. It is a prospective, single-center cohort study of patients with scheduler TAVR procedure by the Heart Team of the Amiens University Hospital. We complete the method section, page 5, line 98, as follows: This is a prospective single-center cohort study involving patients scheduled for TAVR procedures performed by the Heart Team at Amiens University Hospital for severe AS and hospitalized in the Cardiovascular Intensive Care Unit (CICU) at the same hospital.
- If it is prospective (as it appears) but a sample size was not calculated a priori, specify the reason for this and consider calculating the study power post hoc.
Response: We thank the reviewer for this remark. We did not calculate the sample size because we don't have data about RV-LSF in the TAVR population. However, with the data from the study, if we were to conduct this study prospectively, in order to achieve a 15% difference in sensitivity and specificity with a bilateral alpha risk of 0.05% (which is an acceptable limit for comparing diagnostic tests), it would have been necessary to include 179 patients.
We complete the limit section for the reason, page 19, line 358, as follows :
Secondly, our sample size did not allow for an analysis of different RV-PA ratios, including quartiles, and we did not calculate the required number of patients because we did not have data on RV-LSF in this population.
And page 19, line 362, for the post hoc analyze of the sample size :
However, if this study had been conducted prospectively, to achieve a 15% difference in sensitivity and specificity with a 0.05% bilateral alpha risk (which is an acceptable limit for comparing diagnostic tests), it would have been necessary to include 179 patients.
- It seems peculiar that the study's objective was to investigate if the RV-PASP ratio was better than TAPSE/PASP, while all primary and secondary endpoints focused on the RV-PASP relationship and MACE. Comparative data should be included if that is the stated objective.
Response: We thank the reviewer for this remark. A typo has crept into the objective. The main objective was to compare the prognostic effectiveness between the RV-LSF/PASP ratio and the TAPSE/PASP ratio in detecting MACE within a contemporary cohort of TAVR patients. Furthermore, we evaluated the prognostic relevance of both RV-PA coupling parameters in this patient population.
For more clarity, we correct the sentence, page 4, line 90, as follow: Consequently, we compared the prognostic effectiveness between the RV-LSF/PASP ratio and the TAPSE/PASP ratio in detecting MACE within a contemporary cohort of TAVR patients. Furthermore, we evaluated the prognostic relevance of both RV-PA coupling parameters in this patient population.
Results:
- Over 200 patients who underwent TAVR were excluded from the study. What is the reason? This would mean they are not consecutive cases. This exclusion should be acknowledged as a limitation.
Response: We thank the reviewer for this remark. Two hundred seven patients did not meet the inclusion criteria, mainly because another team performed the TAVR procedure. We are the only tertiary hospital center with the presence of a cardiac surgeon in case of complications in our region, and two others performed the TAVR procedure on their patients. We did not include these patients because patient selection and post-procedure follow-up are not conducted at Amiens Hospital, making it challenging to monitor these patients.
We complete the limitation section, page 19, line 352, as follow: Firstly, only TAVR procedures performed by the Amiens Hospital Team were included. Indeed, we are the sole tertiary center with authorization for TAVR procedures, while two other teams conducted TAVR procedures at our center. We excluded 149 patients from these teams because they have their own patient selection and follow-up processes, making it challenging to monitor patients post-procedure and collect data.
- It would be beneficial to include MACE events in the patient flowchart, separated by the type of event within this combined outcome.
Response: We thank the reviewer for this remark. We corrected the flowchart and added the type of event of the MACE criteria.
- I wonder about the remaining parameters of Speckle Tracking, both left and right ventricular, atrial, and their prognostic value. Were these obtained? - Otherwise, it is an interesting and well-structured section.
Response: We appreciate the reviewer's comment. We did not perform other speckle tracking parameters. We believe the reviewer is referring to the global strain of the left ventricle, the RV free wall longitudinal strain (RV-FLWS), and the left and right atrial strains. We did not assess atrial strains (left and right) or the global strain of the left ventricle because our study solely focuses on right ventricular coupling. Regarding RV-FWLS, more than 30% of our patients did not have optimal quality measurements for analyzing the ventricular wall, even with auto-strain.
We added this limitation, page 20, line 377, as follows: The RV-FLWS/PASP ratio is attractive for assessing RV-PA coupling (7). However, our study did not include RV-FLWS in the statistical analysis because more than 20% of patients with an RV-LSF measurement could not obtain a reliable RV-FLWS measurement using Autostrain software.
Discussion:
- Minor correction: Additional space in line 230, 261, and throughout the section, there is incorrect spacing before and after punctuation marks (line 290). Correct this.
Response: We apologize for these typo errors and have corrected them.
- It would be interesting to include other right ventricular parameters that have proven useful in the prognostic evaluation of patients undergoing TAVR. It would be relevant to place the studied parameters in this context.
Response: We thank the reviewer for this comment. We have included RV systolic conventional parameters (TAPSE, S' wave, and RV-FAC), as well as advanced parameters (RV-LSF). We have made changes to Table 3 and the hazard ratios (HR) in the multivariable models. RV-LSF was the only parameter associated with MACE in all three models.
This is in line with the literature, demonstrating that RV-LSF is a superior parameter to conventional ones in identifying patients at risk of MACE because it is more precise in diagnosing patients with right-sided dysfunction.
We complete the discussion section, page 17, line 321, as follows :
In our study, RVsD before TAVR, defined by a RV-LSF <20%, was the only RV systolic function parameter independently associated with MACE at six months. This finding is consistent with several studies demonstrating an association between RVsD, assessed through an RV-LSF <20%, and the occurrence of MACE(9,10). Indeed, RV-LSF seems to be a more precise parameter than conventional in identifying patients with RVsD in various cardiovascular pathologies (8,11). RV-LSF assesses the global RV systolic function, contrary to TAPSE, and likely enables early detection of RVsD (18). In our study, patients did not yet exhibit significant impacts on the RV, including the absence of major tricuspid regurgitation or significant pulmonary hypertension. This is probably because patients received treatment for their AS at an earlier stage, and medical optimization likely helped prevent RV impairment, which is a crucial step in the progression of AS disease.
In limitations, the study's nature, which is not consecutive patients, and the reasons for exclusion should also be included.
Response: We thank the reviewer for this remark and we complete the limit section, page 19, line 352, as follows: Firstly, only TAVR procedures performed by the Amiens Hospital Team were included. Indeed, we are the sole tertiary center with authorization for TAVR procedures, while two other teams conducted TAVR procedures at our center. We excluded 149 patients from these teams because they have their own patient selection and follow-up processes, making it challenging to monitor patients post-procedure and collect data.
Round 2
Reviewer 1 Report
Comments and Suggestions for Authors
I thank the authors for addressing most of my comments of the first review explaining and clarifying the issues mentioned.
Although the authors addressed my comments and clarified most issues listed in the first review I am questioning the value of this study. As the authors state by themselves in response to my comment number 6 ‘We concur that the added value of the RV-LSF/PAPS ratio is low, and its utility is likely limited.’. As shown in the presented data the consideration of this echocardiographic right ventricular marker did not significantly add to the already established and easy to assess parameters BNP and EuroScore II to detect patients at risk for heart failure hospitalization and mortality. Therefore, the authors now state in the subheading ‘Clinical value of RV-PA coupling parameters’ of the discussion that their new parameter is very much limited (lines 308-311) and added several limitations in the respective paragraph of the discussion.
This study leaves us with a new parameter associated with MACE in TAVR without an advantage or easier applicability in daily clinical practice compared to the already widely-established markers BNP and EuroScore II which also are significantly different between the here described cohorts.
Please find some minor comments below mostly addressing style and data presentation which needs to be improved and especially unified in the entire manuscript. These have not been thoroughly addressed in the first review process.
1. Table 2: Please unify the style of the presented data. For example, line ‘Aortic valve area’: Change ‘,’ to ‘.’in No MACE cohort. Also decide if the authors want to present decimal numbers and stay with a consistent number for the median and interquartile range (for example see line ‘E wave’ or ‘mean transaortic gradient’ and many more) for both cohorts in the entire manuscript. The same is true for the presentation of hazard ratios and the confidence intervals in the entire manuscript (for example please see line 235).
2. Referring to my comment 11 of the first review, please stay consistent in presenting p-values in the entire manuscript. I agree with the suggestion of the authors of two decimal numbers for non-significant p-values and three decimal numbers for significant p-values. Please also unify the presentation of p-values using a lowercase non-italic ‘p’ and decide whether to use spaces or not in the entire manuscript.
3. Please erase redundant spaces (for example in the first paragraph auf the results section) and please consistently use for example ether ‘RV-LSF >20%’ (line 299) or ‘RV-LSF < 20%’ (line 237).
4. Please correct RL-LSF/PASP into RV-LSF/PASP and unify the presentation regarding the spaces after ‘< / ≥’ and the unit of the variable.
5. Please clarify the interquartile range of the ‘Aortic peak velocity’ of No MACE cohort as the presented value cannot be true.
6. Please clarify what the authors mean by ‘surgical approach’ for TAVR.
7. Table 1: Please change the phrase ‘Female gender’ to ‘Female sex’ when informing on the biological sex of the patients (unless you want to present the social sex).
8. On my mind: I am curious about the statement of the authors in response to my comment 8 that patients might be in an earlier stage of aortic valve disease when 6-month MACE rate was very much higher than in other studies as stated in the limitations section of the manuscript. In my opinion the study cohort mostly consists of normal-flow high gradient aortic valve stenoses due to the normal ejection fraction, high elevated mean transaortic gradient and aortic peak velocity. This might potentially be an explanation for the normal sized right ventricles with normal systolic function in the absence of tricuspid regurgitation. This might potentially be different in a population of patients suffering from low-flow low-gradient aortic stenoses in context of HFrEF.
Author Response
I thank the authors for addressing most of my comments of the first review explaining and clarifying the issues mentioned.
Although the authors addressed my comments and clarified most issues listed in the first review I am questioning the value of this study. As the authors state by themselves in response to my comment number 6 ‘We concur that the added value of the RV-LSF/PAPS ratio is low, and its utility is likely limited.’. As shown in the presented data the consideration of this echocardiographic right ventricular marker did not significantly add to the already established and easy to assess parameters BNP and EuroScore II to detect patients at risk for heart failure hospitalization and mortality. Therefore, the authors now state in the subheading ‘Clinical value of RV-PA coupling parameters’ of the discussion that their new parameter is very much limited (lines 308-311) and added several limitations in the respective paragraph of the discussion.
This study leaves us with a new parameter associated with MACE in TAVR without an advantage or easier applicability in daily clinical practice compared to the already widely-established markers BNP and EuroScore II which also are significantly different between the here described cohorts.
Response: We appreciate the reviewer's opinion on our article. We agree that our article is not going to revolutionize the management of patients undergoing TAVR and that our new parameter is not a miracle identifier for high-risk MACE patients.
In line with feedback from various reviewers, we have moderated the interpretation of our results. It appears that this parameter still outperforms TAPSE/PASP, a parameter currently enjoying a burgeoning and controversial literature.
Certainly, our results may not be as positive as in some publications, but we believe that it is necessary, for the integrity of scientific research, also to publish studies that do not only have positive results. Indeed, our article merits recognition for introducing a new parameter with an advanced technique and for providing certain results that may contribute to the evaluation of patients undergoing TAVR
Please find some minor comments below mostly addressing style and data presentation which needs to be improved and especially unified in the entire manuscript. These have not been thoroughly addressed in the first review process.
- Table 2: Please unify the style of the presented data. For example, line ‘Aortic valve area’: Change ‘,’ to ‘.’in No MACE cohort.
Also decide if the authors want to present decimal numbers and stay with a consistent number for the median and interquartile range (for example see line ‘E wave’ or ‘mean transaortic gradient’ and many more) for both cohorts in the entire manuscript. The same is true for the presentation of hazard ratios and the confidence intervals in the entire manuscript (for example please see line 235).
- Referring to my comment 11 of the first review, please stay consistent in presenting p-values in the entire manuscript. I agree with the suggestion of the authors of two decimal numbers for non-significant p-values and three decimal numbers for significant p-values. Please also unify the presentation of p-values using a lowercase non-italic ‘p’ and decide whether to use spaces or not in the entire manuscript.
- Please erase redundant spaces (for example in the first paragraph auf the results section) and please consistently use for example ether ‘RV-LSF >20%’ (line 299) or ‘RV-LSF < 20%’ (line 237).
- Please correct RL-LSF/PASP into RV-LSF/PASP and unify the presentation regarding the spaces after ‘< / ≥’ and the unit of the variable.
- Please clarify the interquartile range of the ‘Aortic peak velocity’ of No MACE cohort as the presented value cannot be true.
- Table 1: Please change the phrase ‘Female gender’ to ‘Female sex’ when informing on the biological sex of the patients (unless you want to present the social sex).
Response: We thank the reviewer for this remark. We have corrected and optimized the data presentation by reducing spaces and standardizing the statistical layout.
- Please clarify what the authors mean by ‘surgical approach’ for TAVR.
Response: We thank the reviewer for this comment. We added the meaning of the surgical approach in the Table 2, as follow: The surgical approach was defined by the fact that a surgeon had to perform surgical access at the femoral site in case of a failure to puncture the femoral vessels or had to pre-dilate the vessels before the TAVR team could proceed with their procedure.
- On my mind: I am curious about the statement of the authors in response to my comment 8 that patients might be in an earlier stage of aortic valve disease when 6-month MACE rate was very much higher than in other studies as stated in the limitations section of the manuscript. In my opinion the study cohort mostly consists of normal-flow high gradient aortic valve stenoses due to the normal ejection fraction, high elevated mean transaortic gradient and aortic peak velocity. This might potentially be an explanation for the normal sized right ventricles with normal systolic function in the absence of tricuspid regurgitation. This might potentially be different in a population of patients suffering from low-flow low-gradient aortic stenoses in context of HFrEF.
Response: We thank the reviewer for this remark. Indeed, most of the patients exhibited normal-flow high-gradient severe AS, and the sample size was too small to conduct statistical tests based on the hemodynamic type of AS. As suggested by the reviewer, this type of AS could explain the limited impact on the dilation of the right ventricle.
We added this point, page 17, line 325, as follow : In our study, patients did not exhibit significant impacts on the size of the RV chamber, including the absence of major tricuspid regurgitation or significant pulmonary hypertension. This is probably because most patients presented with normal-flow high-gradient severe AS, received treatment for their AS at an earlier stage, and medical optimization likely helped prevent RV impairment, which is a crucial step in the progression of AS disease.